# Evaluation of a Method for Determining Material Strength Based on Hardness Measurements: A Case Study of the Ti6Al4V Alloy

**DOI:** 10.3390/ma18163726

**Published:** 2025-08-08

**Authors:** Karolina Karolewska, Mateusz Wirwicki, Bogdan Ligaj

**Affiliations:** 1Faculty of Mechanical Engineering, al. PBS Bydgoszcz University of Science and Technology, Kaliskiego 7, 85-796 Bydgoszcz, Poland; 2Faculty of Mechatronics, ul. UKW Kazimierz Wielki University, Kopernika 1, 85-074 Bydgoszcz, Poland; wirwicki@ukw.edu.pl; 3BSW University of Bydgoszcz, ul. Unii Lubelskiej 4C, 85-059 Bydgoszcz, Poland; bogdan.ligaj.1974@gmail.com

**Keywords:** Ti6Al4V alloy, tensile strength, hardness, additive manufacturing, DMLS, heat treatment, mechanical properties, microstructure, non-destructive testing, metal AM

## Abstract

The aim of this study was to evaluate the feasibility of estimating the tensile strength of Ti6Al4V alloy, based on HV measurements. The investigation included samples that were manufactured using both additive technology and conventional methods, under various conditions: as-built, heat-treated, and untested mechanically. Static tensile tests and HV measurements were performed to assess the influence of the manufacturing method, heat treatment, and mechanical loading on material performance. The highest tensile strength was recorded for as-built samples, while the greatest ductility was observed in conventionally drawn bar samples. Hardness values generally correlated with tensile strength trends; however, in heat-treated specimens, the relationship between hardness and tensile strength was found to be nonlinear. Specimens that were not subjected to tensile testing exhibited higher HV values than their mechanically tested counterparts, indicating a potential effect of prior deformation on the local material condition. The results confirm that hardness testing can be a useful indirect method for estimating the tensile strength of Ti6Al4V, particularly in materials with controlled and uniform microstructures. For additively manufactured and heat-treated materials, however, the current empirical models may require adjustment or enhancement using advanced predictive approaches. The proposed indirect method offers an alternative to destructive testing, especially in the industrial quality control context for metal AM.

## 1. Introduction

Determining the mechanical properties of engineering materials, particularly their tensile strength, is one of the key tasks in mechanical engineering. This parameter determines whether a given material can be used in structural components, especially in industries with high safety requirements, such as the aerospace, biomedical, and energy sectors. Traditional testing methods such as the tensile test, while providing precise data, are also destructive techniques that require properly prepared samples and specialized equipment. As a result, their application in routine quality control may be limited, due to cost and time consumption [1,2].

In this context, indirect methods for assessing mechanical properties are gaining increasing interest, with hardness analysis being the most commonly used technique. Hardness (HV), which is defined as a measure of a material’s resistance to local plastic deformation, can be determined relatively quickly and inexpensively, and its measurement is often non-destructive or minimally invasive. Importantly, numerous studies indicate a correlation between hardness and tensile strength, which creates the potential to use hardness as a predictor of the strength of metallic materials [3,4].

One of the materials for which this issue is of particular importance is the titanium alloy Ti6Al4V, which belongs to the group of α+β dual-phase alloys. It is characterized by a high strength-to-weight ratio, excellent corrosion resistance, and outstanding biocompatibility, making it a dominant material in aerospace, implantology, and military applications. At the same time, the mechanical properties of Ti6Al4V are highly dependent on its microstructure, which can be significantly modified by the manufacturing technology used and any subsequent heat treatment processes [5].

In recent years, the additive manufacturing (AM) of metals has been developing rapidly, particularly the direct metal laser sintering (DMLS) method, which belongs to the class of selective laser sintering technologies. This technique enables the tool-free production of geometrically complex components with high dimensional accuracy and efficient material usage. However, structures produced in this way exhibit a microstructure that differs from that of conventionally manufactured materials, including a characteristic grain alignment along the layer-by-layer build direction, the presence of pores, microdefects, and residual stresses. These features have a direct impact on mechanical properties, including hardness and strength [6,7].

Compared to materials produced by traditional methods, such as rolling or drawing, DMLS materials often exhibit greater anisotropy in their properties and a stronger dependence on the build direction. For example, the study by Leuders et al. (2013) demonstrated significant differences in tensile strength and elongation depending on the sample orientation relative to the build axis [8]. For this reason, appropriate post-process heat treatment is crucial in order to homogenize the microstructure, reduce the residual stresses, and eliminate porosity.

Heat treatment, when applied to both DMLS and conventionally produced samples, enables controlled phase transformation and modification of the morphology of the α and β phases. With appropriately selected annealing parameters, a more homogeneous microstructure can be achieved, which translates into stabilized mechanical properties, including improved ductility and fatigue resistance. Another important aspect is the influence of heat treatment on hardness values, which directly affects the accuracy of strength estimation using indirect methods [9].

In this study, four groups of Ti6Al4V alloy samples were used, differing in their manufacturing method and heat treatment condition: DMLS samples in the ‘as-built’ state, DMLS samples after annealing, drawn bar samples without heat treatment, and bar samples after annealing. This variation in sample types makes it possible to evaluate the influence of the microstructure and manufacturing technology on the relationship between hardness and tensile strength.

Modeling the relationship between hardness and tensile strength can take various forms—from simple linear models to more complex power–law or logarithmic functions. The most commonly cited approach is the model proposed by Cahoon et al. (1971), in which tensile strength is expressed as a function of Vickers hardness and the elastic modulus. While this model is widely used, it is based on an assumption of microstructural homogeneity, which may not be applicable to additively manufactured materials [4].

The literature emphasizes that applying universal correlations between hardness and tensile strength carries the risk of prediction errors, particularly for materials with complex microstructures or significant technological differences [10,11]. Therefore, many research teams recommend developing dedicated, empirical relationships tailored to a specific material, processing condition, and hardness range.

The use of hardness as an indicator of material quality holds significant practical importance—especially in the context of industrial additive manufacturing. The ability to non-invasively estimate tensile strength enables faster production decisions, reduces waste, and lowers the costs associated with destructive testing. Therefore, the development of accurate hardness–strength models tailored to the specific characteristics of DMLS and Ti6Al4V can greatly enhance the efficiency of industrial quality control [12].

At the same time, it should be remembered that hardness measurement is not without its limitations. The hardness value may depend on measurement conditions (load, duration, and surface condition), as well as on local structural inhomogeneities. In the case of Ti6Al4V, especially after additive manufacturing processes, significant hardness fluctuations may occur, even within a single sample [13].

Moreover, the homogeneity of the microstructure plays a key role in the accuracy of estimations. Microdefects, chemical segregations, microstructural gradients, and grain orientation can disrupt the relationship between hardness and strength. Therefore, studies in this area should be conducted on samples representing the full range of material conditions and processing technologies.

The Ti6Al4V alloy is also characterized by the presence of two phases (α and β), the volume fraction, morphology, and distribution of which influence both the hardness and strength. Phase transformations during heat treatment, as well as differences between the microstructure of DMLS and conventionally produced materials, can lead to significant variations in the hardness–strength correlation, which should be taken into account when developing predictive models [5].

A review of the literature shows that although there are numerous studies analyzing the relationship between hardness and strength for various metal alloys, data for the Ti6Al4V alloy—especially in the context of DMLS—remain limited. Studies such as those conducted by Kasperovich and Hausmann (2015) and Qiu et al. (2015) are promising but highlight the need for further work on standardizing approaches and validating models under different technological conditions [14,15].

The aim of this article is, therefore, to evaluate the effectiveness of an indirect method for estimating the tensile strength of the Ti6Al4V alloy, based on hardness test results. The study analyzes samples that were produced using the DMLS method, as well as drawn bar stock, both before and after heat treatment. The obtained results will make it possible to verify the suitability of the selected correlation models and to assess the impact of manufacturing technology and microstructure on prediction accuracy.

## 2. Research Object

The research object consisted of samples made of Ti6Al4V ELI alloy intended for biomedical applications, with a lower oxygen content. Table 1 shows the chemical composition of the titanium powder used to produce the test samples.

The test samples were printed on an EOS M290 device. The manufacturing process was characterized by the following parameters: laser power of 400 W, time of the laser beam on the material, 80 µs, hatch distance of 60 µm, layer thickness of 60 µm, and scanning speed of 5000 mm/s. The scanning rotation angle between successive superimposed layers was 0.

Samples with a circular cross-section were adopted for the study of static properties. The geometric and physical forms of the research objects are shown in Figure 1.

Two types of samples made by the DMLS method were accepted for testing. The first series was in the form of as-built samples, without additional heat or mechanical treatment. The second series of samples was subjected to a heat treatment process consisting of the supersaturation of the test material at a temperature of 980 °C for 1 h in a Seco Warwick Case Master Evolution single-chamber vacuum furnace. The third group consisted of samples made of annealed drawn bars. The samples were divided into 6 groups of 3. Table 2 shows the designations of the samples for the respective groups.

The AS, HT, and DB samples were tested under static tensile loads. The ASW, HTW, and DBW samples were not subjected to this analysis, but were instead used to create the samples used in the hardness tests and microscopic examinations. They acted as reference (comparative) samples in determining how thermal processes and the process of testing static properties may affect the hardness of the material.

## 3. Research Methods

### 3.1. Tensile Properties

The static tensile test made it possible to determine the mechanical properties of the material, i.e., tensile strength S_u_, yield point S_y0.2_, Young’s modulus E, and elongation A. The tests, conducted under static load conditions, were carried out on an INSTRON 8502 testing machine (Figure 2). The test was carried out, based on a displacement of the machine piston by 0.05 mm/s. The parameters recorded during the tests were force and displacement. The displacement during the tests was determined using the classic INSTRON 2620-601 extensometer with a measuring base of 8 mm and a measuring range of 1 mm, which was used to determine the deformation value. During the static tensile test, five samples from each group (AS, HT, and DB) were used.

### 3.2. Material Hardness Tests

The material hardness tests were carried out on a Huatec HV_10_ hardness tester (Figure 3). The hardness test was carried out using the Vickers HV5 scale with a force equal to 49.03 N. The AS, ASW, HT, HTW, DB, and DBW samples were tested. Hardness measurements were made on the cross-section of the sample on the *x-y* plane. Figure 4 shows the hardness measurement points for the samples.

### 3.3. Tests of the Microstructure of Materials 

After the tests of mechanical properties and hardness measurements were conducted, the samples were taken for metallographic tests. Microscopic photos were taken along the *z*-axis in two areas: P1—in the grip part of the sample, and P2—in the location of the fracture of the samples subjected to the static tensile test, which is shown schematically in Figure 5.

## 4. Research Results

### 4.1. Tensile Strength Results

Table 3 shows the test results, which were the mean value with any deviations of the following parameters: tensile strength, proof stress, Young’s modulus, and elongation. The results were compared to those obtained for samples made of a drawn bar with the same geometry, as shown in Figure 3.

In this study, a static tensile test was conducted on three types of Ti6Al4V alloy samples. The test results are presented as a stress–strain graph (Figure 6). Analysis of the graph clearly distinguishes the mechanical characteristics of each material.

The AS sample exhibited the highest tensile strength (ultimate tensile strength (UTS) ≈ 1450 MPa), accompanied by relatively low elongation (ε ≈ 8–9%). The stress–strain curve is characterized by a steep rise in the maximum stress, followed by a sudden fracture, which is typical for materials with high strength and limited ductility. The high strength of the AS sample results from its specific columnar microstructure, which is formed during rapid solidification in the DMLS process, as well as the presence of residual stresses and unannealed dislocations [7,8].

The HT sample, obtained by heat-treating the AS sample, showed lower tensile strength (UTS ≈ 1050 MPa) and even lower elongation (ε ≈ 4–5%). This reduction is due to microstructural changes occurring during annealing, including stress relaxation, phase transformations, and a decrease in dislocation density [9]. Despite improved structural homogeneity, the material lost part of its initial strength without a noticeable gain in ductility, which may be the result of suboptimal heat treatment parameters (e.g., an excessively high temperature or insufficient annealing time).

The DB sample (drawn bar, without heat treatment) exhibited the lowest tensile strength (UTS ≈ 950 MPa) but the highest elongation, reaching as much as 26–27%. The DB tensile curve features a long plastic plateau, indicating excellent ductility and a stable deformation mechanism. The results confirm the expected behavior of conventional metallic materials when subjected to cold working processes such as drawing, which promote structural uniformity and increase the number of obstacles to dislocation movement [11].

### 4.2. Analysis of Tensile Strength Results

When analyzing the results of the static properties tests, it can be seen that the samples printed in the as-built condition (AS specimen) are characterized by the highest strength. However, they do not have as much ductility as the samples made of drawn bars (DB specimen), which is also visible in the breakthroughs of the samples after the static tensile test. This material behavior is influenced by the method of producing the material. The printed samples are sintered in each layer, which induces a type of weld to form. This significantly affects not only the high brittleness but also the high strength of elements made with additive technology.

The amount of work that must be applied to damage the sample during the static tensile test was also analyzed. The calculated work is the area under the sample stretching plot, which is shown schematically in Figure 7.

The heat-treated samples did not improve their properties, which finding contrasts with most of the reports in the literature, where heat treatment is generally shown to enhance ductility and homogenize the microstructure of additively manufactured Ti6Al4V components [9,18,19,20]. Compared to the as-built samples, they lost both strength and plasticity. The material may exhibit such behavior due to insufficient heat treatment time and a certain temperature, which is higher than the β-transus transformation temperature. The highest ductility and the uniform α + β structure for the Ti6Al4V material are obtained concurrently by applying heat treatments above the α-transus and below the β-transus temperatures. The applied heat treatment could also insufficiently eliminate the defects arising in the structure, resulting from the method of producing the sample, i.e., unmelted powder particles or gas being used in the working chamber that was trapped in the molten layer.

The analysis showed that the greatest effort was needed to destroy a sample made of a drawn bar—11.19 J (Figure 8). Much less energy was needed to destroy a sample produced in the laser sintering process—6.20 J—which is 55.4% of the energy needed to destroy a sample made of a drawn bar. The heat-treated printed sample is characterized by the smallest area under the tensile diagram, which translates into the smallest amount of work put into damaging it—1.17 J.

### 4.3. Material Hardness Results

The hardness results for the measurements carried out on the *x-y* plane are presented in Table 4 by means of statistical parameters, i.e., the mean, standard deviation, median, and relative standard deviation.

To verify the potential of using hardness as an indirect indicator for assessing the tensile strength of the Ti6Al4V alloy, Vickers hardness (HV) measurements were carried out for six groups of samples: AS, HT, and DB (samples subjected to tensile testing), as well as ASW, HTW, and DBW (samples not subjected to tensile testing). Each hardness value represents the average of 10 independent measurements, through which it was possible to determine not only the mean value but also the standard deviation, median, and relative standard deviation (%RSD).

The highest average hardness was recorded for the ASW sample (412.7 HV), which was produced using the DMLS method in its as-built state, without annealing and without prior mechanical loading. The hardness of the HTW sample (printed and annealed but not subjected to tensile testing) was 356.9 HV, while the DBW sample (drawn bar, also untested) measured 336.7 HV. These values are slightly higher than those of their corresponding tensile-tested counterparts: AS—347.1 HV, HT—392.9 HV, and DB—294.7 HV. This suggests that performing the tensile test may, in some cases, lead to localized modifications in the material’s structure—such as surface work hardening or microscopic deformation—resulting in slight changes (in most cases, reductions) in hardness.

It is worth noting, however, the unusual result observed for the HT sample, which, despite being annealed, shows greater hardness (392.9 HV) than its corresponding untested counterpart, HTW (356.9 HV). This may indicate structural inhomogeneity, local phase transformations, or the influence of micro-deformations that are introduced during the tensile testing process. Possible variations in the specific locations where the hardness measurements were taken on the sample should also be considered.

From the standpoint of result uniformity, the most stable values were obtained for the DBW (1.3% RSD) and DB (2.2% RSD) samples, confirming the high quality and consistency of the microstructure in the material derived from the drawn bar. In contrast, the samples produced using the DMLS method, particularly the HTW and ASW, showed significantly greater result dispersion (%RSD of 5.8% and 5.5%, respectively), which may stem from the inherent anisotropy of additive materials and local differences in density and phase distribution.

A comparison of the hardness results with the tensile strength values enables an analysis of the potential correlations between these properties. The AS sample, characterized by a tensile strength of approximately 1450 MPa, had an average hardness of 347.1 HV, while the DB sample, with the lowest tensile strength (~950 MPa), reached a hardness of 294.7 HV. As expected, higher hardness translated into higher strength, which confirms the validity of using empirical relationships such as the Cahoon model or other classic HV–UTS correlations.

However, the HT sample disrupts this simple correlation. Despite having greater hardness than the AS sample (392.9 HV vs. 347.1 HV), it achieved significantly lower tensile strength (~1050 MPa). This indicates that surface hardness does not fully reflect the strength properties of the heat-treated material. This may result from the fact that heat treatment modifies not only the local hardness but also the internal microstructure, the distribution of α and β phases, dislocation density, and the residual stress distribution—all of which are critical parameters for the material’s behavior under tensile loading.

The analyses of both the mean values and standard deviations lead to the conclusion that hardness can be a useful but insufficient indicator for indirectly estimating the strength of Ti6Al4V materials, particularly in their heat-treated state. For the as-built samples (AS and DB), the HV–UTS relationship shows strong consistency, justifying the use of predictive models in the context of additive manufacturing without annealing or when using conventional materials. However, for heat-treated materials, it is necessary to consider additional structural parameters, which may require extending the correlation model or employing more advanced predictive techniques (e.g., machine learning).

Additionally, the fact that the samples not subjected to tensile testing (ASW, HTW, and DBW) exhibited higher hardness compared to their tensile-tested counterparts suggests the potential influence of mechanical deformation on subsequent hardness measurements—even though the HV test itself is considered minimally invasive. This further reinforces the need to perform hardness measurements in a controlled manner, taking into account the material’s loading history.

Importantly, the samples not subjected to tensile testing (ASW, HTW, and DBW) exhibited higher hardness values compared to their tensile-tested counterparts. At first glance, this appears contrary to the classical phenomenon of strain hardening; however, additional mechanical and structural effects must be considered. During the tensile test, local relaxation of residual stresses and redistribution of the dislocation structure may occur, leading to reduced post-test hardness. The literature suggests that in materials with a martensitic microstructure (such as the α′ phase that is present in the AS samples), these effects can result in strain softening or may even initiate micro-recrystallization within the necking region [15,21] (Brodsky and Thompson, 1984; Qiu et al., 2015). Additionally, local temperature increases during testing (adiabatic heating) may induce micro-annealing [11], and, in additively manufactured structures, partial stress relaxation due to plastic deformation is also possible [14]. As a result, hardness measurements taken after tensile testing may yield lower values, despite the simultaneous occurrence of classical strain hardening. Similar observations for Ti6Al4V were also reported by Bermingham et al. (2011) and Yadollahi and Shamsaei (2017) [9,13].

In summary, the hardness analysis confirms this test’s potential as a supplementary indicator for assessing the mechanical properties of Ti6Al4V. However, its effectiveness depends on technological consistency, microstructural homogeneity, and knowledge of the processing conditions. Only under such conditions is it possible to build reliable and stable predictive models of tensile strength based on hardness measurements.

### 4.4. Determination of Material Strength Based on Hardness Measurement Results

The presented results of the hardness measurements of Ti6Al4V alloy samples made from a drawn bar and the additive method show differences. The DB samples are characterized by the lowest hardness value, which is also reflected in the strength results obtained via the static tensile test. The difference between the specimen under tension (DB specimen) and the as-built sample (DBW specimen) is 42 HV. This difference may indicate changes in the structure of the material resulting from clamping the testing machine’s jaws on the sample during the test, as well as the static tensile test itself. Similar differences are observed in the AS and ASW samples, where the difference is 65.6 HV, and in the HT and HTW samples, where the difference is 36 HV.

The analysis of the results of the hardness tests on the *x–y* plane showed that the ASW and HT samples have the hardest structure, and the difference between them is 19.8 HV. The heat-treated printed samples (HTW) and printed samples without heat treatment (AS) show a similar hardness—the difference is 9.8 HV. The largest value for the dispersion of hardness in the results was observed for the ASW sample, at 22.8 HV.

When analyzing only the hardness results for those specimens taken from the samples not subjected to the static tensile test, where there are no changes in the material caused by tensile forces, it is noted that the ASW sample has the highest hardness, and the DBW has the lowest. The results of the tensile strength tests are similar, with the highest value for the AS sample and the lowest for DB. This highlights the relationship between the hardness and strength of the material.

The performed hardness measurements made it possible to determine the material strength indirectly. To determine the tensile strength in an analytical manner using the results of the hardness measurements, the formula described in Ref. [5] was used, which is presented below:(1)UTS=HC1−n12.5n1−n
where:

H—Vickers hardness value or HV,

C—material factor, and

n—strengthening factor.

The next stage used Equation (1) to carry out analytical calculations and to determine the local tensile strengths for the obtained values of hardness measurements on the *x-y* plane of the sample cross-section. To determine the tensile strength on the basis of the hardness tests, specimens taken from those samples not subjected to the strength test were used and compared with the material strength when determined through a static tensile test.

Figure 9, Figure 10 and Figure 11 show the material strength results (UTS), calculated on the basis of hardness tests for the ASW, HTW, and DBW samples, respectively. The presented graphs show the strength values in terms of points compliant with the hardness measurements (UTS), their average value, and the average strength value obtained based on the static tensile test, together with the confidence intervals. To calculate the UTS, the cyclic hardening factor for the Ti6Al4V alloy with a value of *n* = 0.9 was used, as reported in Ref. [22]. Two extreme values of the C coefficient have been proposed, which, after converting the UTS, are within the assumed confidence interval.

Figure 9 presents the charts for the ASW sample for two values of the material coefficient, C = 1.9 and C = 2.1. Figure 10 shows the graphs for the HTW sample for two values of the material factor, C = 2.3 and C = 2.5. Figure 11 presents the graphs for the DBW sample for two values of the material factor, C = 2.3 and C = 2.6. Table 5, Table 6 and Table 7 show the calculation results of the UTS mean value, standard deviation, median, and relative standard deviation for the ASW, HTW, and DBW samples, respectively.

A comparison of the results of the experimental tests with the results obtained analytically shows that it is possible to use the hardness tests to determine the strength of the material. The presented test results show that the average UTS values determined through the hardness tests were within the accepted confidence intervals, as determined on the basis of the results of the static tensile test. The different values of the C coefficient are due to the effect of the relevant treatment that was applied and the resulting variable material structure. In each of the presented cases, two extreme values of C were given, between which the strength values are close to the experimental average value.

### 4.5. Analysis of the Material Microstructure Test

Figure 12 shows photos of the individual microstructures of samples made of the Ti6Al4V alloy. Figure 12a,b show the microstructure of a printed sample in the as-built condition, subjected to strength tests (AS specimen). Figure 12c,d show the microstructure of a printed sample, subjected to heat treatment and after the tensile test (HT specimen). Figure 12e,f show the microstructure of a sample made of a pulled bar after testing its mechanical properties (DB specimen). Figure 12g shows the structure of the printed sample in the as-built condition, without undergoing the tensile test (ASW specimen). Figure 12h shows the microstructure before the strength tests of a heat-treated printed sample (HTW specimen). Figure 12a,c,e,g,h show the microstructure in the gripping part (P1). Finally, Figure 12b,d,f show the microstructure in the measuring part (P2).

The AS specimen is characterized by a clearly visible martensitic α′ phase structure with a strongly acicular (needle-like) morphology. The grains are elongated in the *z*-axis direction (the layer-building direction), indicating directional solidification during processing. The α′ martensite forms as a result of the very rapid cooling that is inherent to the additive manufacturing process and the absence of any subsequent heat treatment. This microstructure is typical for as-built 3D-printed materials and is responsible not only for their very high hardness and tensile strength but also for their brittleness and limited ductility. The observed fractures in the tensile-tested sample are consistent with this interpretation—the material exhibits no significant plastic deformation before failure.

In the case of the HT specimen, a more balanced mixture of α (matrix) and β (darker regions) phases is visible, with grains that are less elongated and more equiaxed. The reduced presence of acicular martensitic α′ indicates its decomposition as a result of thermal exposure. Although the microstructure transformed toward a more ductile form, the mechanical properties did not improve. This may be due to an excessive heat treatment temperature (above the β-transus) or an insufficient holding time, which led to incomplete stabilization of the α + β phases and degradation of the microstructure.

The DB specimen exhibits a homogeneous, fine-grained α + β structure that is typical of plastically worked Ti6Al4V products. The α phase appears in plate-like or lamellar form, uniformly distributed within the β matrix. The absence of a grain orientation suggests isotropic mechanical behavior. This microstructure provides well-balanced mechanical properties, combining good strength and ductility. Its uniformity forms a reliable basis for predictive modeling and serves as a reference point for additively manufactured materials.

In the ASW specimen, which is microstructurally identical to the AS specimen but has not been subjected to mechanical deformation, an intense presence of martensitic α′ is also observed, with the grain structure aligned along the layer-building direction (*z*-axis). The absence of deformation marks or residual stress indicators suggests an ‘undisturbed’ material condition post-printing. This sample enables the evaluation of the isolated effect of tensile testing. When compared to the AS specimen, the microstructural differences are minimal, indicating that neither the HV test nor the tensile test significantly alters the deep internal structure. However, both may influence the residual stress state and local deformation characteristics (e.g., dislocation density).

The HTW specimen, which was printed and heat-treated similarly to the HT specimen but was not subjected to tensile loading, exhibits a visibly α + β phase structure composed of larger, less well-defined grains compared to the HT specimen. The absence of mechanical deformation (tensile test) likely resulted in a lack of dynamic grain refinement. The presence of a coarser microstructure may explain both the lower repeatability of hardness measurements and a potential decrease in mechanical performance. This specimen suggests that heat treatment alone may not be sufficient to optimize the material properties—controlled cooling or additional mechanical processing steps might be necessary.

Table 8 presents a comparative summary of the specimens based on their resulting microstructures. The as-built additively manufactured specimens (AS and ASW) exhibit a characteristic martensitic α′ structure, which is responsible for high hardness and strength, but also contributes to significant brittleness. Heat treatment (HT and HTW) leads to decomposition of the martensite and the emergence of a more classical α + β phase mixture; however, the effectiveness of this transformation depends strongly on the heat treatment parameters. In the present study, the applied thermal processing did not significantly improve the mechanical properties. The specimens made from drawn bar material (DB and DBW) demonstrate the most predictable and balanced microstructure, serving as a reference state for comparison.

The lack of a noticeable influence of tensile testing on the macroscopic microstructure of the samples indicates that the dominant factors modifying the internal structure are the manufacturing method (additive vs. conventional) and the applied heat treatment, rather than mechanical deformation during testing.

Both the AS specimen (after tensile testing) and the ASW specimen (untested) represent the characteristic microstructure that is formed by the DMLS (direct metal laser sintering) process. A common feature of these samples is the presence of the martensitic α′ phase, forming a dense, acicular (needle-like) structure with a columnar arrangement aligned along the *z*-axis—the direction of layer-wise additive manufacturing.

The α′ martensite forms as a result of extremely rapid cooling during the metal printing process. It is a metastable phase that is hard and brittle. The directional grain alignment leads to anisotropy in its mechanical properties. The lack of observable microstructural differences between the AS and ASW specimens suggests that tensile loading does not significantly alter the overall grain morphology, although it may locally affect the stress distribution and dislocation density.

The application of heat treatment in the HT and HTW specimens caused significant changes in the material’s structure. The transformation of the α′ martensite into a classical biphasic α + β microstructure is evident. The α grains exhibit a more rounded (spheroidal) morphology, while the β phase appears between them in the intergranular regions.

Although this structure is typical of annealed titanium alloys, the analyzed samples do not display full microstructural homogeneity, and the grains remain relatively coarse. The HT sample, which underwent tensile testing, shows slightly more refined grains compared to the HTW, suggesting that tensile deformation may have promoted partial microstructural reorganization or activated dynamic recrystallization processes.

Despite the phase transformation, the mechanical properties of the HT and HTW specimens deteriorated compared to the as-built condition, which could be attributed to non-optimal annealing parameters, for example, an excessively high temperature leading to overgrowth of the β phase.

The DB specimen (after tensile testing) represents a conventional microstructure that is typical of plastically worked materials. It features a fine lamellar α + β structure, uniformly distributed across the sample cross-section. The absence of preferential grain orientation indicates that the material is isotropic, which results in consistent and predictable mechanical behavior.

Comparison of the microstructures in the P1 (grip section) and P2 (gauge section, fracture zone) regions for selected Ti6Al4V specimens enables an assessment of the influence of tensile loading on local structural changes in the material.

In the AS specimen (additively manufactured, without heat treatment), both P1 and P2 reveal a distinct columnar martensitic α′ microstructure that is characteristic of DMLS-fabricated materials in the as-built state. However, in region P2—the fracture zone—this microstructure shows local disturbances: the acicular α′ phase is partially deformed and disrupted, likely due to tensile stresses and local thermal effects during fracture. Subtle changes in needle orientation and density suggest an increase in dislocation density and the initiation of microcracks. These features indicate that although the overall phase composition remains unchanged, the tensile test introduces localized structural reorganization in the fracture zone.

In the HT specimen (additively manufactured and heat-treated), region P1 displays a balanced α + β microstructure with relatively uniform, spheroidal grains that are typical of annealed titanium alloys. In contrast, region P2 exhibits greater irregularity, with disrupted phase boundaries and a more scattered grain morphology. These changes may indicate grain boundary motion or intragranular deformation induced by tensile loading. In both the AS and HT specimens, it is evident that while tensile testing does not cause phase transformation, it does affect the arrangement and integrity of the microstructure—particularly in DMLS materials, where anisotropy and residual stresses are already present, due to the layer-wise manufacturing process.

For the DB specimen (drawn bar, no heat treatment, and subjected to tensile testing), the differences between P1 and P2 are significantly less pronounced than in the printed samples. Region P1 shows a classic fine lamellar α + β structure that is uniformly distributed and with no preferential grain orientation, confirming the isotropy and homogeneity of conventionally processed Ti6Al4V. In region P2 (the fracture zone), the same general structure is preserved, although the local elongation of α lamellae and slight blurring of phase boundaries can be observed—likely a result of tensile-induced strain and localized stress concentration.

Compared to the printed specimens, the microstructural changes in region P2 of the DB sample are more subtle, confirming the greater structural resistance of the conventional material to tensile deformation. The drawn bar material exhibits high microstructural stability and resistance to localized damage, which translates into more favorable mechanical properties and reduced scatter in terms of both the hardness and tensile strength results.

Microstructural analysis of the Ti6Al4V alloy specimens revealed significant differences resulting from the manufacturing method, the heat treatment, and the presence of mechanical deformation. Samples produced via additive manufacturing (DMLS) in the as-built condition were characterized by a needle-like, columnar martensitic α′ structure, which promotes high hardness and strength but limits ductility and microstructural uniformity. The application of heat treatment led to a transformation into a biphasic α + β structure; however, no significant improvement in mechanical properties was observed, indicating the need for optimization of the annealing parameters.

In contrast, the microstructures of specimens prepared from the drawn bar material exhibited the highest degree of homogeneity and stability, with a fine-grained α + β structure that is typical of conventionally processed titanium alloys, offering a favorable balance between strength and ductility. The effect of tensile deformation on the microstructure was marginal, primarily involving localized strain-induced modifications.

These findings confirm the critical role of microstructures in shaping the mechanical performance of Ti6Al4V and highlight the importance of carefully selecting the manufacturing method and heat treatment parameters in the context of additive technologies for titanium alloys.

## 5. Discussion

The results of our research confirm the effectiveness of the indirect method for estimating ultimate tensile strength (UTS) based on the Vickers hardness (HV) value in Ti6Al4V alloy. However, the degree of agreement with the model largely depends on the manufacturing technology and the structural state of the material. Samples printed using the DMLS method in the as-built condition (AS) exhibited the highest strength and high levels of hardness, corroborating the findings of Gong et al. [7], Vrancken et al. [23], and Tammas-Williams et al. [24], who indicated the presence of a martensitic α′ structure and grain orientation as the main factors contributing to increased hardness.

For those samples subjected to heat treatment (HT), surprisingly lower UTS and elongation values were obtained compared to the as-built state. This result diverges from the literature reports by Bermingham et al. [9] and Ahmed and Rack [25], wherein annealing below the β-transus temperature led to improved ductility while maintaining moderate strength. One potential cause is the use of an excessively high temperature, resulting in grain coarsening and weakening of the phase boundaries.

Samples made from drawn bar (DB) material exhibited the lowest hardness and strength but the highest ductility. This observation aligns with the results of Semiatin et al. [26], Dieter [11], and Facchini et al. [27], who emphasized that conventional plastic deformation yields a lamellar α+β microstructure and isotropic mechanical properties. At the same time, this confirms the effectiveness of HV-UTS models for materials with a homogeneous structure.

It was observed that samples not subjected to mechanical testing (ASW, HTW, and DBW) had higher average HV values than their tensile-tested counterparts. A similar trend was noted by Leuders et al. [8] and Maskery et al. [28], suggesting that static loading may cause internal stress relaxation or local dislocation movement, thereby affecting the HV.

The analytically determined values (using the Cahoon model) for ASW, HTW, and DBW, after accounting for factors C and n, were consistent with the empirical results. A similar approach was employed by Tammas-Williams et al. [24] and Prashanth et al. [12], who achieved high agreement in HV-UTS models for additively manufactured Ti6Al4V structures.

Discrepancies between hardness and strength in the case of the HT samples indicate that the microstructure (particularly a heterogeneous β phase) has a greater influence than surface HV measurements. This is supported by studies from Frazier [29] and Ghidini et al. [18], which suggest the need to account for texture, chemical segregation, and residual stresses when developing predictive models.

The literature, including the work of Yadollahi and Shamsaei [13], indicates that the implementation of machine learning models can accommodate data irregularities and complex structural variables, potentially increasing prediction accuracy for heat-treated materials. This concept is gaining traction, as seen in publications by Garmestani et al. [30] and Tjong and Ma [31], wherein artificial intelligence supports the modeling of metal properties.

Microscopic analysis confirmed the presence of typical martensitic α′ structures in the AS/ASW and α+β in the DB/DBW. Similar observations were made by Vilaro et al. [32] and Zhang et al. [33], emphasizing that the microstructure resulting from DMLS is characterized by strong anisotropy, which influences the fracture mechanics and HV result dispersion.

Ultimately, this study confirms the validity of using HV measurements to assess the quality of Ti6Al4V alloy, although there are still limitations stemming from microstructural complexity. It is necessary to introduce corrections to HV-UTS models for heat-treated cases or to apply hybrid methods that combine structural and mechanical data, as proposed by Liu et al. [34] and Yan et al. [35].

## 6. Conclusions

This study evaluated the feasibility of indirectly determining the tensile strength of Ti6Al4V alloy based on Vickers hardness measurements. The investigation was carried out on specimens produced using both the additive manufacturing method (DMLS) and drawn bar stock, in both the as-built condition and after heat treatment. Some specimens were subjected to tensile testing, while others were used for hardness measurements and microstructural analysis.

The results confirmed that a clear correlation exists between hardness and tensile strength; however, this relationship is strongly dependent on the manufacturing method and the structural state of the specimens. The highest tensile strength was achieved by the as-built DMLS specimens (AS specimens), which featured a characteristic columnar martensitic α′ structure. However, these specimens exhibited significantly reduced ductility, as evidenced by their low elongation values and brittle fracture mode. The heat-treated specimens (HT specimens), despite their expected improvement in properties, showed a decrease in both strength and ductility. This is attributed to the application of an excessively high annealing temperature (above the β-transus) and the potentially ineffective elimination of structural defects.

In contrast, conventionally manufactured specimens (DB specimens) exhibited the lowest tensile strength but the highest ductility and the most homogeneous microstructure. The resulting fine-grained α + β structure, with no preferential grain orientation, provided stable and isotropic mechanical properties, making this material a reliable reference for further hardness–strength correlation studies.

The hardness measurements revealed that those specimens not subjected to tensile testing (ASW, HTW, and DBW) demonstrated higher average HV values than their tensile-tested counterparts. This may suggest the influence of mechanical testing on the local condition of the material—for example, surface work hardening or relaxation of the dislocation network. The average hardness values obtained for the ASW, HTW, and DBW samples were used to analytically estimate the tensile strength, using established models and selected material coefficients. These calculated values fell within the confidence intervals determined from mechanical testing, confirming the usefulness of the indirect estimation method.

Based on their analysis, it was found that there is a relationship between the strength and hardness of the material and the selection of appropriate material factors. It was found that the strength of a structural element that is produced with additive technology can be determined without needing to use destructive tests, which is economically justified. The analytically obtained results for material strength were comparable with the results obtained for the static tensile test (they fell within the determined confidence interval). For samples printed without heat treatment and not subjected to the static tensile test (ASW) using the proposed analytical method, two material factors were determined: C = 1.9 and C = 2.1. For printed samples that were not subjected to the static tensile test (HTW), the material factors were C = 2.3 and C = 2.5. DBW samples were characterized by the coefficients C = 2.3 and C = 2.6. All analyzed strength results were within the confidence limits.

It should be emphasized, however, that the application of universal hardness–strength correlation models (HV–UTS) may lead to significant predictive errors, especially with heat-treated additively manufactured materials, where the microstructure tends to be heterogeneous and the deformation mechanisms are complex. The variability in hardness values and the limited improvement in mechanical properties after annealing highlight the need to carefully select the heat treatment parameters—particularly temperature and dwell time—ideally keeping them below the β-transus.

Microstructural observations confirmed that the dominant factors affecting both the microstructure and mechanical performance are the manufacturing method and thermal processing. Tensile testing induced only localized changes (e.g., in the fracture zone) without substantially altering the overall phase composition. Therefore, when predicting the tensile strength from hardness data, greater emphasis should be placed on the sample’s structural features and processing history than on its mechanical load history.

In summary, the results demonstrate that hardness can serve as an effective indirect indicator for estimating the tensile strength of Ti6Al4V, especially under controlled conditions and in samples with uniform microstructures (e.g., in the as-built state or after plastic deformation). For heat-treated or structurally complex materials, the accuracy of such estimations may be limited and may require the calibration of local material coefficients or the application of more advanced models (e.g., machine learning-based approaches). Despite these limitations, the proposed method represents a viable alternative to conventional destructive testing, supporting efficient quality control in industrial applications—particularly in the additive manufacturing of titanium alloys.

## Figures and Tables

**Figure 1 materials-18-03726-f001:**
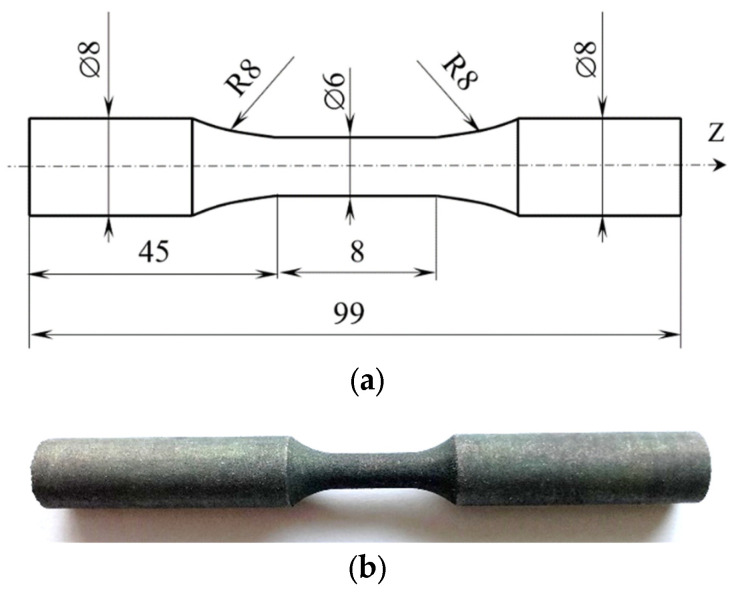
Research material: (**a**) sample diagram with dimensions; (**b**) physical form of the sample. Data from [17].

**Figure 2 materials-18-03726-f002:**
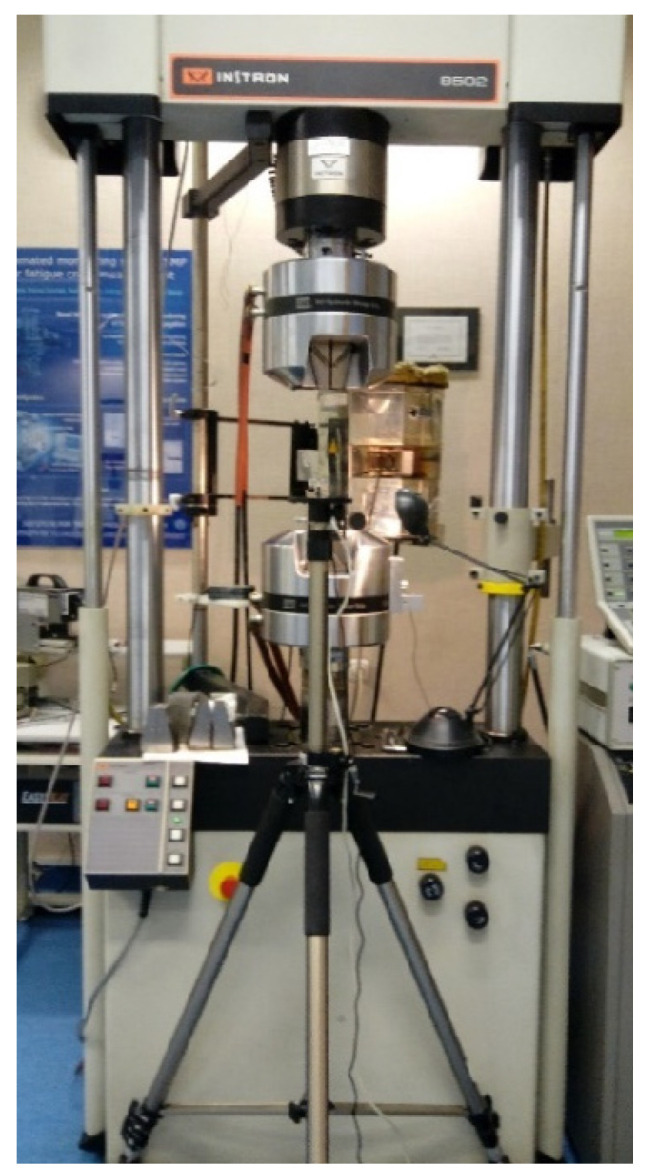
Tensile strength machine – INSTRON 8502, Bydgoszcz, POLAND.

**Figure 3 materials-18-03726-f003:**
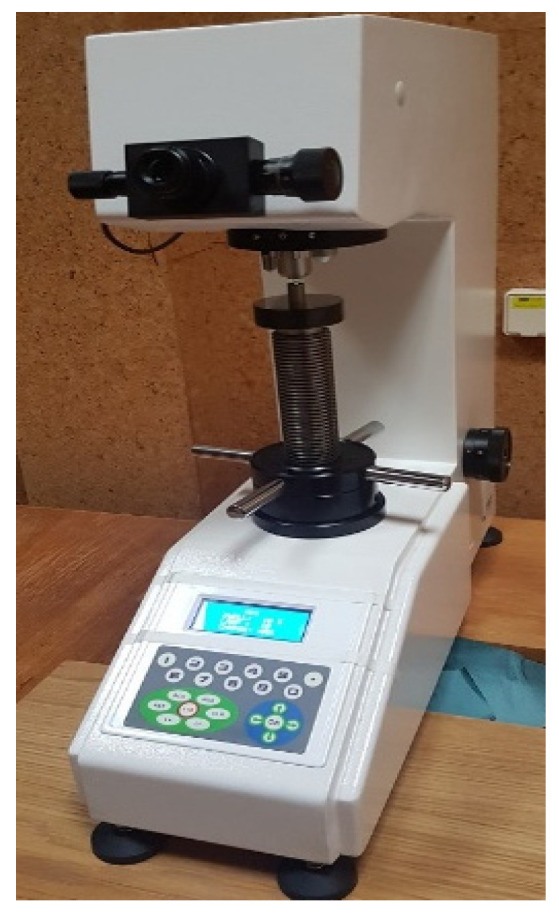
HUATEC HV10 hardness tester, Bydgoszcz, POLAND.

**Figure 4 materials-18-03726-f004:**
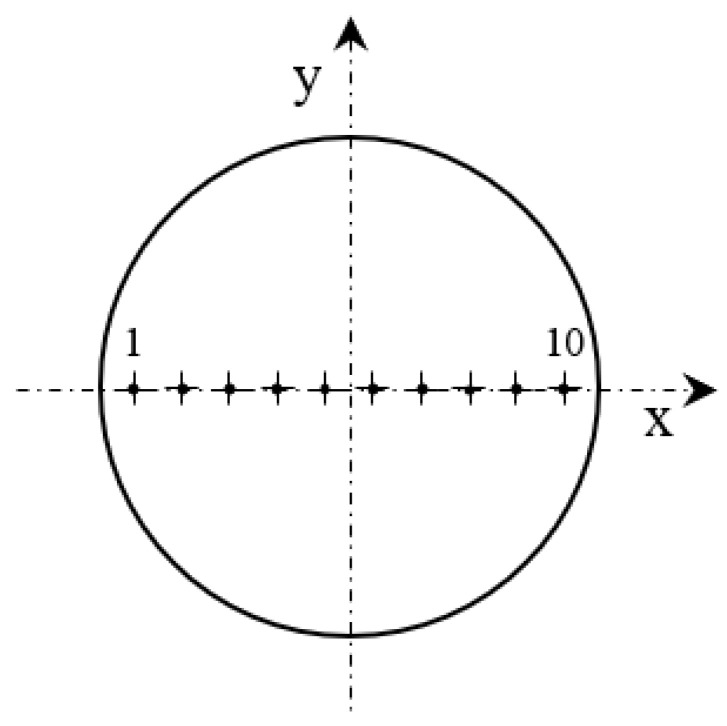
Schematic representation of the hardness measurement method on the *x-y* plane.

**Figure 5 materials-18-03726-f005:**
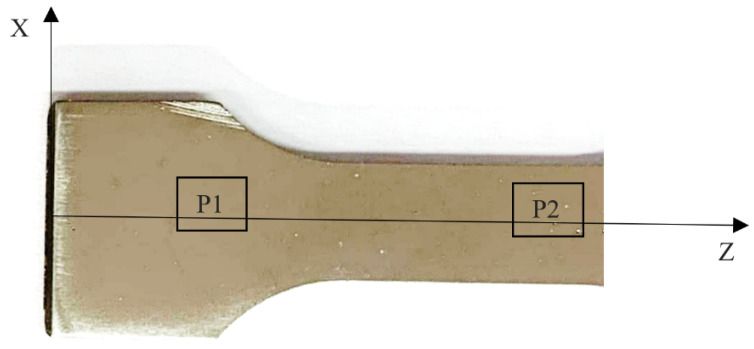
Scheme of the sample used for metallographic tests, with the places for taking microstructural pictures marked.

**Figure 6 materials-18-03726-f006:**
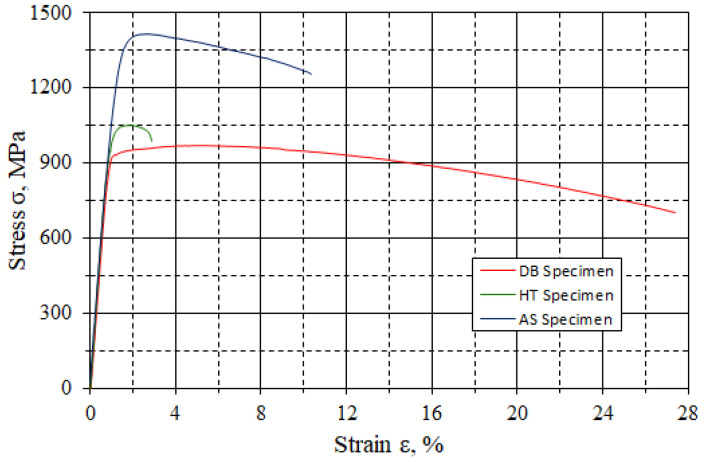
Sample plot of static tension results.

**Figure 7 materials-18-03726-f007:**
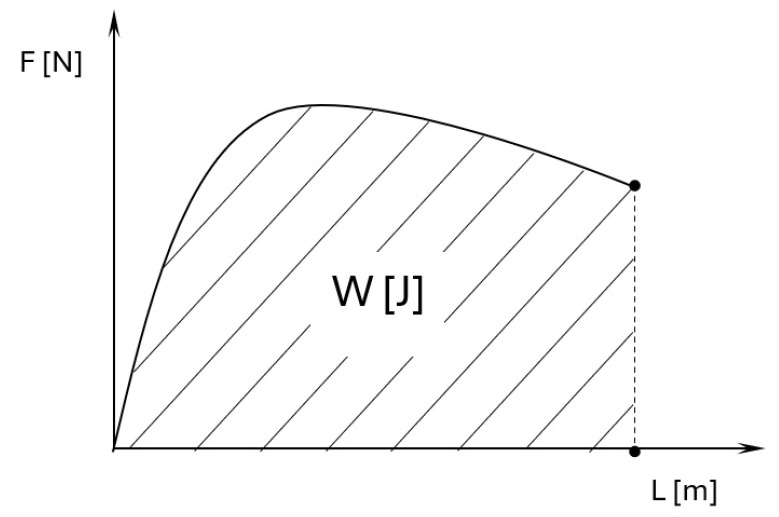
Scheme for determining the energy needed to destroy samples during strength tests.

**Figure 8 materials-18-03726-f008:**
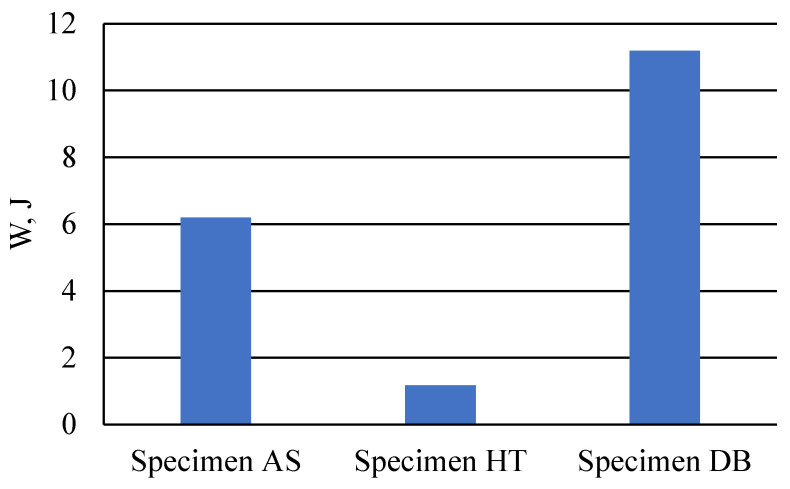
Graph showing the amount of energy needed to damage the samples.

**Figure 9 materials-18-03726-f009:**
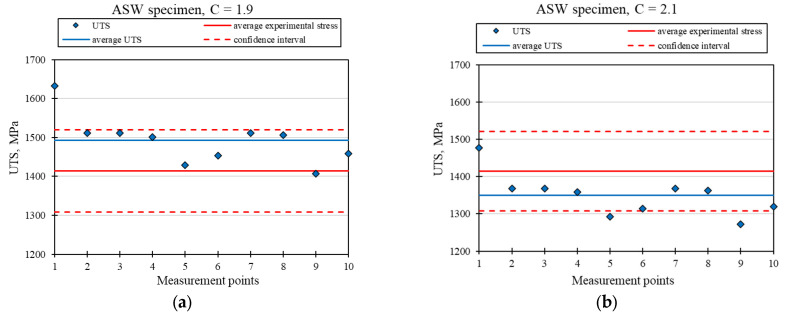
Graph of the calculated UTS for the ASW sample with the adopted value of the material factor at: (**a**) C = 1.9, (**b**) C = 2.1.

**Figure 10 materials-18-03726-f010:**
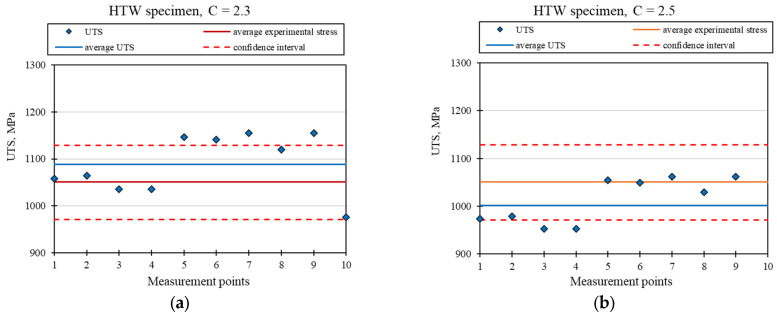
Graph of the calculated UTS for the HTW sample with the adopted value of the material factor at: (**a**) C = 2.3, (**b**) C = 2.5.

**Figure 11 materials-18-03726-f011:**
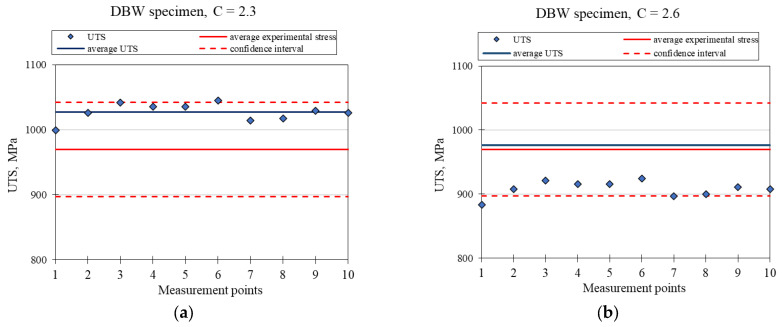
Graph of the calculated UTS for the DBW sample with the adopted value of the material factor at: (**a**) C = 2.3, (**b**) C = 2.6.

**Figure 12 materials-18-03726-f012:**
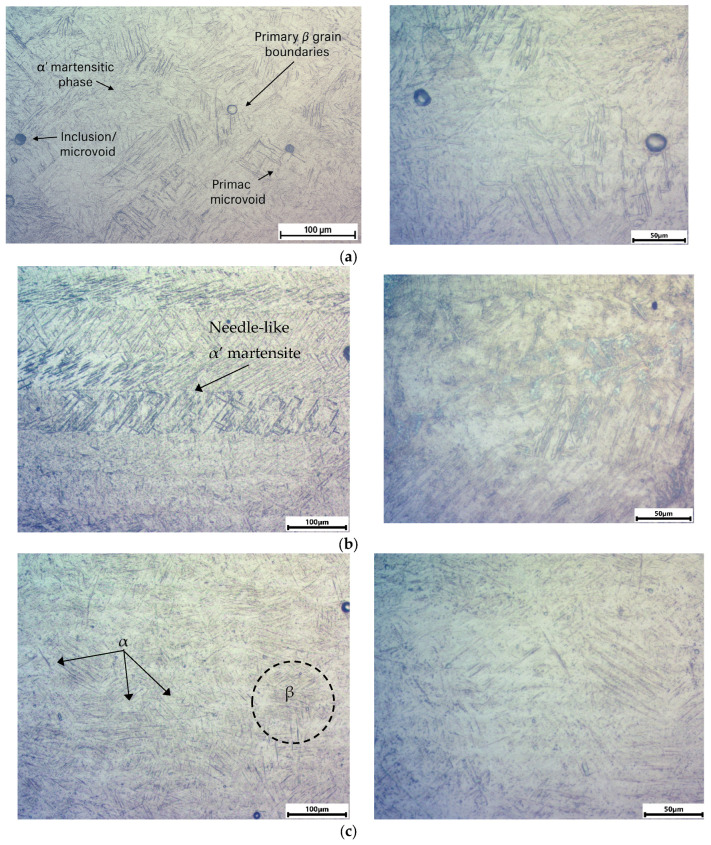
Microstructure pictures of the Ti6Al4V alloy samples on the left and specific area on the right: (**a**) AS specimen—P1, (**b**) AS specimen—P2, (**c**) HT specimen—P1, (**d**) HT specimen—P2, (**e**) DB specimen—P1, (**f**) DB specimen—P2, (**g**) ASW specimen—P1, and (**h**) HTW specimen—P1.

**Table 1 materials-18-03726-t001:** Ti6Al4V ELI powder chemistry according to Renishaw’s catalog [16], given as a weight percentage (wt.%).

Al	V	Fe	O	C	N	H	Y	Other	Ti
5.5–6.5	3.5–4.5	≤0.25	≤0.13	≤0.08	≤0.05	≤0.012	≤0.005	≤0.4	rest

**Table 2 materials-18-03726-t002:** Description and marking of samples taken for testing.

No.	Group Name	Description
1	AS Specimen	A printed sample without heat treatment, subjected to a static tensile test
2	HT Specimen	A printed sample with heat treatment, subjected to a static tensile test
3	DB Specimen	A sample from a drawn bar without heat treatment, subjected to a static tensile test
4	ASW Specimen	A printed sample without heat treatment, not subjected to a static tensile test
5	HTW Specimen	A printed sample with heat treatment, not subjected to a static tensile test
6	DBW Specimen	A sample made of a drawn bar without heat treatment, not subjected to the static tensile test

**Table 3 materials-18-03726-t003:** Results of the tests of static tensile strength.

No.	Specimen No	S_u_	S_y0.2_	E	A
		MPa	MPa	MPa	%
1	2	3	4	5	6
1	AS specimen	1414.3 ± 106.1	1278 ± 95.8	106,300 ± 7972.5	10.4 ± 0.8
2	HT specimen	1050.5 ± 78.8	992 ± 74.4	117,840 ± 8838	2.9 ± 0.2
3	DB specimen	969.7 ± 72.8	927 ± 69.5	112,000 ± 8400	27.4 ± 2.1

**Table 4 materials-18-03726-t004:** Analysis of hardness measurement results on the *x–y* plane.

Specimen	Number of Measurements	Average [hv]	Standard Deviation [hv]	Median [hv]	Relative Standard Deviation [%]
AS Specimen	10	347.1	12.5	343.5	3.6
HT Specimen	10	392.9	15,8	394.0	4.0
DB Specimen	10	294.7	6.4	294.5	2.2
ASW Specimen	10	412.7	22.8	410.3	5.5
HTW Specimen	10	356.9	20.6	358.0	5.8
DBW Specimen	10	336.7	4.5	337.0	1.3

**Table 5 materials-18-03726-t005:** Analysis of the results of the analytical calculations of tensile strength for the ASW sample.

Material Factor	Number of Measurements	UTS Average [MPa]	Standard Deviation [MPa]	Median [MPa]	Relative Standard Deviation [%]
1.9	10	1492.2	62.3	1503.6	4.2
2.1	10	1350.1	56.4	1360.4	4.2

**Table 6 materials-18-03726-t006:** Analysis of the calculation results of the tensile strength analysis for the HTW sample.

Material Factor	Number of Measurements	UTS Average [MPa]	Standard Deviation [MPa]	Median [MPa]	Relative Standard Deviation [%]
2.3	10	1088.6	62.8	1091.9	5.7
2.5	10	1001.5	57.8	1004.6	5.7

**Table 7 materials-18-03726-t007:** Analysis of the results of analytical calculations of tensile strength for the DBW sample.

Material Factor	Number of Measurements	UTS Average [MPa]	Standard Deviation [MPa]	Median [MPa]	Relative Standard Deviation [%]
2.3	10	1026.9	13.8	1027.9	1.3
2.6	10	908.5	12.2	909.3	1.3

**Table 8 materials-18-03726-t008:** Microstructural and mechanical characteristics of titanium alloy specimens.

No.	Specimen	Technology	Heat Treatment	Tensile Test State	Microstructure Characteristics	Dominant Phase Features	Impact on Mechanical Properties
1	AS Specimen	DMLS (3D printing)	None	After tensile testing	Martensitic α′, needle-like, columnar, Z-orientation	High hardness, brittleness	Very high strength, low ductility
2	HT Specimen	DMLS	Yes	After tensile testing	α + β, spheroidal, decomposed martensite	Coarser grains, partial transformation	Reduced strength and ductility compared to AS
3	DB Specimen	Drawn bar	None	After tensile testing	α + β, fine-grained, lamellar, homogeneous	Dominant α phase, regular	Good balance of strength and ductility
4	ASW Specimen	DMLS	None	Without tensile testing	Martensitic α′, same as AS, Z-orientation	Similar to AS, no deformation	Baseline for evaluating the effect of tensile testing
5	HTW Specimen	DMLS	Yes	Without tensile testing	α + β, larger grains than in HT, less ordered	Weaker cohesion, less homogeneous structure	Potentially lower hardness, variable strength

## Data Availability

The original contributions presented in this study are included in the article material. Further inquiries can be directed to the corresponding author.

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
