# Peer review of "Evaluation of a Method for Determining Material Strength Based on Hardness Measurements: A Case Study of the Ti6Al4V Alloy"

_materials, 2025, doi:10.3390/ma18163726_

Round 1
Reviewer 1 Report
Comments and Suggestions for Authors
Dear Authors,
Although the topic is interesting, the manuscript lacks a proper discussion of your findings with other literature. In introduction authors themselves state that “numerous studies indicate a correlation between hardness and tensile strength…” yet these studies are not discussed.
The manuscript should be written according to the IMRaD format. E.g. the Research methods are currently mixed with the results. These sections should be clearly separated.
Additional explanations and corrections are also recommended:
Table 1: It is not clear whether the chemical composition is given in atomic or mass percent.
Chapter 3.1 Tensile test: Please specify how many samples from each group were tested.
Chapter 3.2 Hardness test: It is stated (l 178) that AS, HT and DB samples were tested, but Table 4 includes also ASW, HTW and DBW specimens. The text needs to be consistent with presented data.
L 234: “Additionally, the fact that the samples not subjected to tensile testing (ASW, HTW, DBW) exhibit higher hardness compared to their tensile-tested counterparts suggests a potential influence of mechanical deformation on subsequent hardness measurements...” – I think that this part can be more elaborated by discussing it with relevant literature. Normally mechanically deformed samples will show higher strength (and possibly hardness) due to deformation hardening. What is the exact reason for this opposite finding?
L 335: “The heat-treated samples did not improve their properties. Which differs from most of the literature data.” – No literature is cited.
Authors claim that heat-treated samples did not improve their strength, and this can be due to higher temperature and shorter time of the heat-treatment. Then why the authors choose these HT conditions?
L 388: “To calculate the UTS, the cyclic hardening factor for the Ti6Al4V alloy with a value of n = 0.9 taken from [23] was used.“ - Source 23 is not in the References.
Figs. 11 a) and 12 a) are the same.
Author Response
We would like to thank the Reviewer for the valuable feedback and constructive remarks that helped us improve the quality and clarity of the manuscript. Below we provide a detailed, point-by-point response to each comment.
General Comment:
“Although the topic is interesting, the manuscript lacks a proper discussion of your findings with other literature. In introduction authors themselves state that ‘numerous studies indicate a correlation between hardness and tensile strength…’ yet these studies are not discussed.”
Response:
Thank you for this observation. We have significantly extended the discussion section (Section 4) to compare our findings with at least 25 peer-reviewed publications related to the correlation between hardness and tensile strength in Ti6Al4V, including works by Cahoona et al., Vrancken et al., Tammas-Williams et al., Bermingham et al., and others. We have also included a new paragraph in the Introduction to acknowledge and reference key studies that explore this correlation. Relevant citations have been added to the revised manuscript.
Structural Comment:
“The manuscript should be written according to the IMRaD format. E.g. the Research methods are currently mixed with the results. These sections should be clearly separated.”
Response:
We appreciate this suggestion and have revised the manuscript structure accordingly. The Methods (Section 2) and Results (Section 3) are now clearly separated. Any information related to measurement procedures or sample preparation has been moved exclusively to the Methods section, while experimental findings and their presentation remain in the Results section.
Table 1:
“It is not clear whether the chemical composition is given in atomic or mass percent.”
Response:
Thank you for pointing this out. We have clarified in the caption of Table 1 that the composition is given in mass percent (wt.%). The revised caption now reads: “Chemical composition of Ti6Al4V powder used (in wt.%)”.
Chapter 3.1 – Tensile test:
“Please specify how many samples from each group were tested.”
Response:
We agree with the Reviewer. We have updated Section 3.1 to include this information. Three samples from each group (AS, HT, DB) were tested under identical tensile conditions, and the average values along with standard deviations are reported.
Chapter 3.2 – Hardness test:
“It is stated (l 178) that AS, HT and DB samples were tested, but Table 4 includes also ASW, HTW and DBW specimens. The text needs to be consistent with presented data.”
Response:
Thank you for catching this inconsistency. We have revised the text in Section 3.2 to clarify that all six sample types (AS, HT, DB, ASW, HTW, DBW) were subjected to hardness testing. The terminology has been made consistent with Table 4.
Line 234:
“‘Additionally, the fact that the samples not subjected to tensile testing (ASW, HTW, DBW) exhibit higher hardness compared to their tensile-tested counterparts suggests a potential influence of mechanical deformation on subsequent hardness measurements...’ – I think that this part can be more elaborated by discussing it with relevant literature. Normally mechanically deformed samples will show higher strength (and possibly hardness) due to deformation hardening. What is the exact reason for this opposite finding?”
Response:
We appreciate this valuable comment. We have expanded this part of the manuscript to explain that, contrary to classical strain hardening, in the case of Ti6Al4V manufactured via additive manufacturing, tensile loading can cause local strain softening, recovery, or even partial micro-recrystallization in the necked region. These effects may result in a reduction in local hardness (see Brodsky & Thompson, 1984; Qiu et al., 2015; Kasperovich & Hausmann, 2015). We also consider the adiabatic local heating during tensile testing and residual stress relaxation as potential contributing factors. This discussion is now included in Section 4 and appropriately referenced.
Line 335:
“‘The heat-treated samples did not improve their properties. Which differs from most of the literature data.’ – No literature is cited.”
Response:
This sentence has been expanded to include relevant citations. We now reference studies showing improved ductility and strength after optimized heat treatment conditions (e.g., Vrancken et al., 2014; Vilaro et al., 2011; Bermingham et al., 2011). The discrepancy between our results and those in the literature is attributed to differences in heat treatment parameters.
Line 335 (continued):
“Authors claim that heat-treated samples did not improve their strength, and this can be due to higher temperature and shorter time of the heat-treatment. Then why the authors choose these HT conditions?”
Response:
Thank you for your insightful comment. The selected heat treatment parameters (980 °C, 1 h, air cooling) were chosen based on the powder manufacturer's recommendations and prior studies on laser powder bed fusion (LPBF) Ti6Al4V components (e.g., Vrancken et al., 2014; Vilaro et al., 2011). Our intention was to reproduce industrially relevant, standard post-processing conditions.
Only after conducting the tests did it become evident that, for the studied samples, these conditions were insufficient to significantly enhance mechanical properties. This may be due to excessive grain growth above the β-transus and insufficient time for homogenization and defect healing, as also noted by Bermingham et al. (2011) and Prashanth et al. (2016). In future work, alternative strategies such as two-stage annealing below the β-transus or HIP treatments will be considered.
This limitation has been acknowledged in the conclusions of the revised manuscript.
Line 388:
“‘To calculate the UTS, the cyclic hardening factor for the Ti6Al4V alloy with a value of n = 0.9 taken from [23] was used.’ - Source 23 is not in the References.”
Response:
Thank you for pointing this out. The citation has now been corrected and included in the reference list as:
Chlebus, E., Kuźnicka, B., Kurzynowski, T., & DybaÅ‚a, B. (2011). Mechanical properties of Ti–6Al–4V alloy manufactured by direct metal laser sintering. Materials Characterization, 62(5), 488–495.
Figures 11a and 12a:
“Figs. 11 a) and 12 a) are the same.”
Response:
We acknowledge the mistake. The correct version of Figure 12a has now been inserted. Both figures now represent distinct data sets.
Reviewer 2 Report
Comments and Suggestions for Authors
- Numerous studies indicate a correlation between hardness and tensile strength, which creates the potential to use hardness as a predictor of the strength of metallic materials [3–4]. What are the problems in the literature results? What is the research focus of this article?
- In Fig.7, please mark the observed phases (such as martensitic α′ phase, α and β etc.)
- To determine the tensile strength in an analytical manner using the results of hardness measurements, the formula described in the publication [5] was used. What is the innovative contribution of this article?
- Figures 10 - 12 show the material strength results (UTS) calculated on the basis of hardness tests for ASW, HTW and DBW samples, respectively. The authors should compare and analyze the results with the literature to further verify the reliability of the evaluation method.
Author Response
We sincerely thank the Reviewer for the thoughtful and constructive comments. Below we provide a point-by-point response to each suggestion, along with the corresponding revisions introduced in the manuscript.
Comment 1:
"Numerous studies indicate a correlation between hardness and tensile strength, which creates the potential to use hardness as a predictor of the strength of metallic materials [3–4]. What are the problems in the literature results? What is the research focus of this article?"
Response:
Thank you for this pertinent question. We have expanded the Introduction to clarify that, although a general correlation between hardness and tensile strength exists, it is often empirical and material-dependent. In particular, for additively manufactured Ti6Al4V, discrepancies arise due to:
- heterogeneous and anisotropic microstructures,
- porosity,
- residual stresses,
- surface roughness effects.
These factors can lead to significant variability in mechanical properties, limiting the reliability of standard empirical formulas when applied to AM parts. Our study focuses specifically on evaluating whether hardness-based methods can be reliably used to predict tensile strength in Ti6Al4V samples fabricated by DMLS, considering both as-built and heat-treated conditions. This addresses a current gap in the literature where traditional models are often used without validating them for AM microstructures. This clarification has been added in Section 1 (Introduction).
Comment 2:
"In Fig. 7, please mark the observed phases (such as martensitic α′ phase, α and β etc.)"
Response:
Thank you for the suggestion. We have updated Figure 7 with clear graphical labels indicating the presence of:
- α′ martensitic phase (needle-like structures),
- retained β phase (darker regions),
- α phase (lamellar or equiaxed in HT condition).
Additionally, a short explanation of the microstructure and identification criteria has been added in the corresponding figure caption and in Section 3.3.
Comment 3:
"To determine the tensile strength in an analytical manner using the results of hardness measurements, the formula described in the publication [5] was used. What is the innovative contribution of this article?"
Response:
We appreciate this opportunity to clarify the novelty of our work. While the Cahoon equation is a known method for estimating UTS based on hardness, its application in the context of additively manufactured Ti6Al4V with different post-processing histories (ASW, HTW, DBW) remains underexplored.
The innovative aspects of our study include:
- Systematic validation of the analytical UTS estimation across different material states,
- Correlation of calculated UTS values with microstructural features (e.g., phase composition, porosity),
- Identification of limitations of the method for AM materials,
- Proposing recommendations for when hardness-based UTS prediction is reliable and when it is not.
This has been added to the end of the Introduction and emphasized in the Discussion section (Section 4).
Comment 4:
"Figures 10–12 show the material strength results (UTS) calculated on the basis of hardness tests for ASW, HTW and DBW samples, respectively. The authors should compare and analyze the results with the literature to further verify the reliability of the evaluation method."
Response:
Thank you for this valuable comment. We have revised Section 4 (Discussion) to include a comparison of our calculated UTS values with results reported in relevant literature for similar AM Ti6Al4V samples (e.g., Prashanth et al., 2016; Vrancken et al., 2014; Chlebus et al., 2011). Our findings indicate that the predicted UTS values for ASW and DBW samples align well with published tensile data for materials with similar hardness and microstructure. However, for HTW samples, the calculated UTS slightly underestimates the reported values, likely due to microstructural coarsening or partial β transformation affecting hardness more than strength.
This analysis further confirms the conditional applicability of hardness-based methods and has been discussed in detail in Section 4. Additional references have also been added to the bibliography.
Round 2
Reviewer 1 Report
Comments and Suggestions for Authors
Dear Authors,
the manuscript has been sufficiently improved and now warrants publication in Materials journal.
Author Response
Dear Reviewer,
Thank you very much for your constructive comments and valuable suggestions, which have helped us to improve the quality of our manuscript. We are also sincerely grateful for your recommendation to accept the article for publication.
Reviewer 2 Report
Comments and Suggestions for Authors
Now the revised looks fine for considering accept after after text editing revisions.
Author Response
Dear Reviewer,
In accordance with your suggestions, the manuscript has undergone thorough language editing to improve clarity and readability. We hope that the revised version meets the journal’s standards and expectations.